# Heme Peroxidases at Unperturbed and Inflamed Mucous Surfaces

**DOI:** 10.3390/antiox10111805

**Published:** 2021-11-12

**Authors:** Jürgen Arnhold

**Affiliations:** Medical Faculty, Institute of Medical Physics and Biophysics, Leipzig University, 04107 Leipzig, Germany; juergen.arnhold@medizin.uni-leipzig.de

**Keywords:** lactoperoxidase, myeloperoxidase, eosinophils peroxidase, mucous surfaces, inflammation, hypothiocyanite, hypochlorous acid, cystic fibrosis, allergies

## Abstract

In our organism, mucous surfaces are important boundaries against the environmental milieu with defined fluxes of metabolites through these surfaces and specific rules for defense reactions. Major mucous surfaces are formed by epithelia of the respiratory system and the digestive tract. The heme peroxidases lactoperoxidase (LPO), myeloperoxidase (MPO), and eosinophil peroxidase (EPO) contribute to immune protection at epithelial surfaces and in secretions. Whereas LPO is secreted from epithelial cells and maintains microbes in surface linings on low level, MPO and EPO are released from recruited neutrophils and eosinophils, respectively, at inflamed mucous surfaces. Activated heme peroxidases are able to oxidize (pseudo)halides to hypohalous acids and hypothiocyanite. These products are involved in the defense against pathogens, but can also contribute to cell and tissue damage under pathological conditions. This review highlights the beneficial and harmful functions of LPO, MPO, and EPO at unperturbed and inflamed mucous surfaces. Among the disorders, special attention is directed to cystic fibrosis and allergic reactions.

## 1. Introduction

Our organism is permanently exposed to various kinds of microorganisms. With the predominant majority of these microorganisms, we are living in peaceful co-existence. The immune system controls the activities of bacteria, viruses, fungi, protozoa, parasites, etc., as well as the responses to any threat by the recruitment and activation of immune cells and the formation of antibodies [1]. In addition, proteins of the acute phase, complement, coagulation, and contact systems can be activated and closely assist the immune response [2]. These activities are directed to decrease tissue damage and to restore homeostasis of the affected tissues. Nevertheless, local or systemic inflammations can culminate with pronounced tissue destruction and pathological states in severe cases. The division of microorganisms into innocuous or harmful ones highly depends on the current immunological state of the individuals. It is well known that usually harmless microbes can provoke serious deteriorations in immunosuppressed patients [3,4].

Mucous surfaces form specialized, indispensable to life compartments in the organism with their own specific rules for defense reactions [5]. Important internal surfaces are the epithelia of the upper respiratory system and the lungs (bronchia, bronchioles, and alveola), and surfaces of the digestive tract (epithelia of oral cavity, esophagus, stomach, small intestine, and colon). In addition, several glands expel their secretions at these and other surface areas. Maintaining the integrity of mucous surfaces is highly essential for the long-term functioning and survival of the organism. On the one hand, these epithelia form a physical barrier. On the other hand, they are embedded in multiple functional processes with defined fluxes of certain metabolites through these surfaces.

Heme peroxidases are known to contribute to immune protection in humans and many other organisms. In mucous linings and secretions, lactoperoxidase (LPO) is an essential component to maintain pathogen contamination on low level [6]. Myeloperoxidase (MPO) is another heme peroxidase that is enriched in polymorphonuclear leukocytes, also known as neutrophils [7,8]. At inflamed sites, these cells recognize, phagocytose, and degrade microbes and fungi under the involvement of MPO. A third peroxidase, the eosinophil peroxidase (EPO) contributes to killing larger pathogens such as helminths by eosinophil granulocytes (eosinophils) [9]. Under inflammatory conditions, immune cell-derived heme peroxidases are also discussed for their involvement in destructive reactions. In particular, MPO is known to exert a dual role as a beneficial or damaging agent in immune reactions [10].

This review highlights the participation of heme peroxidases in immune reactions at unperturbed and inflamed mucous surfaces and the involvement of peroxidase-derived agents in the pathogenesis of diseases of mucous epithelia.

## 2. Organization of Immune Defense at Mucous Surfaces

At mucous surfaces, the immune defense is directed to ensure the physical integrity of the epithelium and to protect adjacent tissues from any threat without disturbing specific metabolic exchange processes within and through the epithelial layer. Two major activities of epithelial cells contribute to immune protection: (i) Gel-forming mucins and an arsenal of antimicrobial agents, including LPO and thiocyanate, are permanently secreted by epithelial cells into the luminal space, and (ii) activation and damage of epithelial cells can induce immune defense reactions.

### 2.1. Secretion of Immune-Protective Agents

Different immune-protecting agents are secreted by the epithelial cells of mucous surfaces (Figure 1). These agents are directed to maintain homeostatic conditions in the surface epithelia and adjacent luminal space.

The epithelia of inner surfaces are covered by a thin mucous layer, in which the main components are produced and released by epithelial Goblet cells and mucous cells in submucosal glands [11]. This mucous lining consists of polymeric glycoproteins, the mucins that form a network with viscoelastic and lubricious properties. In mucins, oligosaccharide units are linked to a polypeptide backbone. Non-glycosylated regions of mucin chains enable interchain disulfide crosslinking and are vulnerable to proteolysis [12,13]. Mucin molecules and rheology of the mucous layer differ at various mucosal epithelia in dependence on their key physiological functions [14,15].

Intact mucin networks protect the underlying cell layer from dryness, allow the selective permeability of gases and nutrients, and represent a physical barrier that keeps commensal microorganisms and pathogens away from the epithelial surface [5,14,16]. Usually, the mucous lining is free of or contains only a few pathogens.

Airway epithelia are covered by a periciliary and a mucous layer. The first layer is responsible for lubrication of ciliary beat, transport of the mucous layer, and represents a size exclusion barrier. In the mucous layer, mucins form a viscoelastic, isotonic fluid. Both layers act together in mucociliary elimination of inhaled pathogens [12,16].

In the lungs, there is an intense cross-talk between epithelial cells and alveolar macrophages to maintain homeostasis [17]. Alveolar macrophages are maintained in a quiescent state by neighbored epithelial cells under non-activating conditions [18]. This regulatory interaction is lost due to epithelial damage or by the presence of pathogens. Lung epithelial cells are able to secrete a wide variety of antimicrobial agents such as β-defensins, lactoferrin, LL-37, lysozyme, and secretory leukocyte proteinase inhibitor. In addition, they can produce various surfactant proteins that are involved in pathogen clearance [19]. Macrophages phagocytose undergoing cells, cell debris, and pathogens and contribute to the regulation of the inflammatory process [20].

In the mucous lining, additional protection is given against potential invaders by bactericidal components such as resistin-like molecule β and zymogen granulae protein 16 that are secreted from Goblet cells [21,22], as well as by antifungal peptides such as histatins and MUC7 12-mer [23,24]. In the small intestine, Paneth cells release bactericidal defensins, lysozyme, and cryptdins into the gut lumen [25].

In addition to these directly acting microbicidal agents, components of the lactoperoxidase-H_2_O_2_-thiocyanate system are secreted at mucous epithelia. As outlined in more detail in Section 4, this system exerts immune protection in mucous linings and secretions by synthesis of the antimicrobial agent hypothiocyanite/hypothiocyanous acid.

Mucous epithelial cells secrete glutathione, which is a major protective agent against oxidants. Particular high levels of glutathione are found in airway lining fluid [26]. In lung epithelial fluid, other abundant species with antioxidative properties are ascorbate and uric acid [27]. Lactoferrin is another protective component at mucous surfaces. It exhibits antibacterial activity against Gram-negative bacteria and prevents bacterial growth by sequestration of free iron ions [28]. Moreover, the iron-binding protein transferrin is enriched in human bronchoalveolar fluid [29].

### 2.2. Immune-Activating Mechanisms at Mucous Surfaces

While secretory processes from epithelial cells are usually sufficient to warrant homeostatic conditions in the surrounding luminal space, the activation of the host’s immune system is important when any molecular patterns are sensed by pattern recognition receptors in the epithelium. Under inflammatory conditions, both pathogen-associated molecular patterns (PAMPs) and damage-associated molecular patterns (DAMPs, also known as alarmins) initiate the recruitment and activation of immune cells to the inflamed area [30,31,32].

Immune response depends on the pathogen type and the degree of cell and tissue destruction. According to the dominating pathogens, three major routes for immune response are differentiated [33]. The type 1 immune response with monocytes as effector cells is directed to protect cells and tissues against intracellular bacteria, protozoa, and viruses. Helminths and other extracellular parasites are targeted by basophils, mast cells, and eosinophils in the type 2 immune response. In a third route, neutrophils are mainly involved to recognize and kill extracellular bacteria and fungi. These pathways are characterized by their own set of secreted cytokines and immunoglobulins, as well as by the development of typical disease scenarios in the case of dysregulation. A general problem in the thorough description of alterations at diseased mucous surfaces results from the fact that several routes of immune response can concomitantly be activated.

Although all of the three types of immune response can play a role at inflamed mucous surfaces, the attention is often focused on type 2 immune reactions as parasites, helminths, and viruses target predominantly mucous epithelial surfaces. Stressed and damaged epithelial cells release several molecules playing the role of DAMPs and initiators of allergic inflammations. Among them are thymic stromal lymphopoietin (TSLP), IL-33, and, IL-25 [34,35,36]. These epithelial-derived cytokines regulate a broad spectrum of immune cell responses and are involved most of all in the induction of Th2 cells, production of IgE by B-lymphocytes, elicitation and activation of mast cells, eosinophils, and basophils [37].

Concerning heme peroxidases, LPO is commonly available in secretions also under normal, non-inflammatory conditions. The other two peroxidases, MPO and EPO, can be found at inflamed areas of mucous surfaces when neutrophils and eosinophils are additionally recruited and activated.

## 3. Important Properties and Reactions of Heme Peroxidases

### 3.1. Selected Structural Properties

Whereas LPO and EPO are monomeric proteins [38,39], MPO has a dimeric structure, where two identical subunits are linked by a disulfide bridge and each subunit consists of a light and heavy polypeptide chain [40]. At neutral pH values, all of the three heme peroxidases are cationic proteins. The isoelectric point of LPO is about 9.5 [41]. The corresponding values for the two other peroxidases are usually given as >10 for MPO [42] or >11 in the case of EPO [43]. Myeloperoxidase is rich in surface-located lysine and arginine residues, which are important for the interaction with negatively charged proteins, glycosaminoglycans, and DNA [44].

Common to all of the three mentioned peroxidases, MPO, EPO, and LPO, is the presence of a heme group (ferric protoporphyrin IX) in each monomeric unit that is coupled in contrast to other heme proteins (such as hemoglobin or cytochromes) by two (LPO, EPO) or three (MPO) covalent linkages to the apoprotein, as shown by X-ray data (MPO, LPO) and biochemical analysis (EPO) [39,45,46]. As a result, the heme becomes more or less bow-shaped, the heme iron is marginally shifted from its central position to the proximal side, electrons of the porphyrin ring are additionally stressed, and the position of the Soret band is redshifted. The strongest effects on heme are observed in the case of MPO. These heme deteriorations provide the basis for the extraordinary reactivity of heme peroxidases including the oxidation of (pseudo)halides and the oxidation of numerous small molecules. In Table 1, selected structural peculiarities of MPO, EPO, and LPO are summarized.

### 3.2. Formation of Compound I

Depending on heme iron valency and the presence of a radical moiety, several redox states of heme peroxidases can be differentiated. In the resting enzyme, heme iron is in the ferric state. Upon reaction with hydrogen peroxide (H_2_O_2_), the so-called Compound I is formed from the ferric enzyme. In Compound I, oxygen is coupled to ferryl iron by a double bond and a further electron is taken from the porphyrin ring that represents a porphyryl cation radical [50]. The highly reactive Compound I is able to catalyze two- and one-electron oxidations of numerous small substrates. In the case of LPO Compound I, the radical moiety of the porphyrin ring can spontaneously be transferred to the apoprotein, most likely a tyrosine residue, forming an apoprotein radical that is known as Compound I* [51,52]. This radical transfer is unknown for MPO and EPO.

### 3.3. Halogenation Cycle

The important substrates for the two-electron oxidation of Compound I of MPO, EPO, and LPO are (pseudo)halides. Their oxidation yields hypohalous acids or hypothiocyanite (OSCN^−^) as major products. Upon these two-electron oxidations, Compound I is reconverted into the ferric enzyme form. The sequence of the two reactions (ferric enzyme → Compound I → ferric enzyme) is called the halogenation cycle (Figure 2).

The three heme peroxidases differ in their ability to oxidize (pseudo)halides. At neutral pH values, only MPO oxidizes Cl^−^ at a reasonable rate [47]. Bromide is well oxidized by MPO and EPO [47,48] and to a minor degree by LPO [51]. Oxidation of I^−^ and thiocyanate (SCN^−^) is known for all of the three heme peroxidases [47,48,51]. Lactoperoxidase Compound I* is unable to oxidize halides and thiocyanate. With increasing acidity, rate constants increase for (pseudo)halide oxidation by MPO and EPO [47,48]. Moreover, at pH 5, EPO is able to oxidize Cl^−^ [48]. Furthermore, the ability of LPO to oxidize SCN^−^ increases under slightly acidic conditions [49].

### 3.4. Peroxidase Cycle

In one-electron oxidations of substrates by Compound I of these peroxidases (and also by Compound I* of LPO) Compound II is formed, which contains an oxo-ferryl heme iron, but no radical moiety neither in the porphyrin ring nor the apoprotein [40]. The list of potential one-electron substrates of Compound I is long. The important one-electron substrates for MPO are selected polyphenols, urate, tyrosine, tryptophan, sulfhydryls, indole derivatives, nitrogen oxide, nitrite, H_2_O_2_, and superoxide anion radicals [53,54,55,56,57,58].

The reduction of Compound II to the ferric enzyme is also coupled with the one-electron oxidation of substrates. In contrast to Compound I, the ability of Compound II for substrate oxidation is restricted to a limited number of substrates. As a result of this restriction, heme peroxidases can accumulate as inactive Compound II in the absence of substrates that are well oxidized by Compound II. For MPO and LPO, efficient substrates for Compound II are superoxide anion radicals, urate, tyrosine, serotonin, nitrite, and selected flavonoids [53,57,58,59,60,61,62,63]. In the presence of these substrates, an accumulation of Compound II can be avoided and the halogenation activity can be enhanced.

The reaction sequence (ferric enzyme → Compound I → Compound II → ferric enzyme) is known as the peroxidase cycle (Figure 2). In the case of LPO, the conversion of Compound I to Compound II can also occur via Compound I*, which is formed from Compound I by an isoelectronic transition.

## 4. The Lactoperoxidase-Hydrogen Peroxide-Thiocyanate System in Mucous Fluids and Secretions

### 4.1. Distribution of LPO

Lactoperoxidase is synthesized in the epithelial cells of secretory surfaces and secreted into the luminal space. This protein is found in the epithelial lining covering the upper airways, and in secretions such as milk, tears, and saliva [64]. In the gut, LPO is expressed in the mouse epithelium, but not in human epithelial cells [65]. Evidently, in the human small intestine and distal colon, the complex immunological defense is ensured without the participation of LPO.

### 4.2. The LPO Knockout Mouse

In mutant mice, the total knockout of the LPO gene causes complex multisystem inflammatory pathology as assessed by histological examination [66], namely inflammation of myocardium, coronary artery, aorta, and cardiac valves. Moreover, other pathologies concern inflammatory airway disease, glomerulonephritis, inflammation in the digestive system, and the development of tumors in different organs [66]. LPO knockout mice have a limited lifetime. About 30% of these mice died before they reached 1 year of age or needed to be killed for human endpoints. Both overweight (and even obese) as well as underweight mice were observed in the LPO knockout cohort [66].

These data indicate that LPO is an important component of immunological control at mucous surfaces and in secretions that can only partly be compensated by other defense mechanisms.

### 4.3. Formation of Hypothiocyanite/Hypothiocyanous Acid

At mucous surfaces and in secretions, LPO is known to act together with H_2_O_2_ and SCN^−^. These components are shortly referred to as the LPO-H_2_O_2_-SCN^−^ system. Two major aspects are under discussion about the mode of action of this system in anti-inflammatory defense at these loci: (i) The formation of the microbicidal hypothiocyanite/hypothiocyanous acid; and (ii) the control over the H_2_O_2_ level.

In the presence of low micromolar levels of H_2_O_2_, ferric LPO is converted into Compound I that oxidizes SCN^−^ to OSCN^−^ in a very rapid reaction [51]. The latter ion is in equilibrium with its protonated form, the hypothiocyanous acid (HOSCN) with a *p*-value of 5.3 [67] or 4.85 [68]. In contrast to HOCl [69,70] and HOBr [71], the reactivity of OSCN^−^/HOSCN is more specific. It predominantly reacts with targets containing accessible sulfhydryl groups or selenocysteine residues [72,73]. Moreover, the uncharged HOSCN is able to permeate through membranes and thus, can enter intracellular compartments. In microorganisms, intracellular glutathione and sulfhydryls in cytosolic enzymes are pronounced targets for oxidized SCN^−^ [74,75]. In secretions, HOSCN can even penetrate into biofilms [74,75].

In epithelial cells, cytosolic and mitochondrial thioredoxin reductases are known to metabolize HOSCN to SCN^−^ and H_2_O and thus, protect these cells from side reactions of HOSCN [76]. Bacteria are unable to inactivate HOSCN in this way [76]. With these activities, the LPO-H_2_O_2_-SCN^−^ system contributes to the control over microorganisms at numerous mucous surfaces and in secretions.

### 4.4. Secretion of Thiocyanate

Thiocyanate is also secreted from epithelial cells of mucous surfaces and secretory glands. In circulating blood, SCN^−^ concentration is about 10–120 µM [77,78]. The blood level of SCN^−^ depends on the patient’s smoking habit and diet regime. Higher SCN^−^ serum concentrations are found in persons smoking heavily or with a strong cabbage diet. In secretory epithelial cells, SCN^−^ is enriched from capillary blood by an active transport mechanism via the sodium-iodide symporter [79]. This transporter, well known from thyroid epithelium, accumulates both SCN^−^ and I^−^ in epithelial cells. Moreover, iodide is well oxidized by LPO Compound I under the formation of an arsenal of oxidized iodine species [80]. However, these products play only a minor role in mucosal defense due to the low abundance of I^−^ in the blood. The plasma level of I^−^ is below 100 nM [81].

Secretion of SCN^−^ (and also I^−^) from epithelial cells at the apical site occurs through several mechanisms [82]. The main focus is usually directed on two anion channels, cystic fibrosis trans-membrane conductance regulator (CFTR) and pendrin. The transport of SCN^−^ through CFTR is stimulated by cAMP, whereas pendrin is sensitive to interleukin-4 [83]. A third mechanism of SCN^−^ transport is mediated by Ca^2^^+^-dependent Cl^−^ channels [83].

In saliva, SCN^−^ concentrations are around 0.5–4 mM [84,85], while the I^−^ value is reported to be 5–22 µM [84]. High micromolar SCN^−^ concentrations were found in tears (150 µM [86]), nasal airway fluid (300–450 µM [87]), and lung airway fluid (270–650 µM [88]).

### 4.5. Sources for Hydrogen Peroxide

There are several sources for H_2_O_2_ in mucous linings. Duox1 and Duox2 are assumed to be the main sources of H_2_O_2_ in the mucous lining. These enzymes are expressed in the apical plasma membrane of airway epithelial cells [89,90,91]. They use electrons from NADPH to reduce dioxygen in order to superoxide anion radicals, which dismutate spontaneously or are catalyzed by superoxide dismutases to H_2_O_2_ and O_2_.

Xanthine oxidase is an additional enzyme that reduces O_2_ to superoxide anion radicals and H_2_O_2_ [92]. Enhanced activities of xanthine oxidase were observed in inflammatory airways epithelia [93,94].

Few bacteria, which can colonize at mucous surfaces, are known to produce H_2_O_2_ [95], namely *Streptococcus pneumoniae* and a few other *Streptococcus* species [96,97]. Microbial H_2_O_2_ inhibits inflammasome-dependent processes of innate immune defense and thus, promotes bacterial colonization [97].

### 4.6. Control over Hydrogen Peroxide

The second major aspect of the mode of action of the LPO-H_2_O_2_-SCN^−^ system concerns the control over the H_2_O_2_ level by LPO. The increased H_2_O_2_ level is assumed to favor oxidative stress and numerous oxidative damage reactions of biological components most probably via the Fenton reaction.

In SCN^−^ oxidation, LPO cycles permanently between the ferric form and Compound I and thus, utilizes 1 mol H_2_O_2_ per 1 mol oxidized SCN^−^. This (pseudo)halogenation cycle can be abated at low SCN^−^ level in mucous linings. Under these conditions, the probability rises for the spontaneous transformation of Compound I into Compound I* [51]. In addition, a broad range of substrates can be oxidized by Compound I (and also by Compound I*) by abstracting one electron under formation of Compound II [52]. In the absence of substrates that are able to reduce Compound II to ferric LPO, this can lead to the arrest of the enzyme as inactive Compound II. As a result of low external SCN^−^, the formation of microbicidal OSCN^−^/HOSCN is diminished and the level of H_2_O_2_ rises.

Hydrogen peroxide is freely permeable through biological membranes. In cells, the level of H_2_O_2_ is controlled by several H_2_O_2_ consuming enzymes such as peroxiredoxins, catalase, and glutathione peroxidase [98,99,100]. In mucous linings and secretions, LPO is a major agent controlling H_2_O_2_. In the airway lining fluid, additional control of the H_2_O_2_ level is carried out by the high yield of glutathione together with extracellular glutathione peroxidase [26,101].

Hydrogen peroxide reacts in the so-called Fenton reaction with transition metal ions (Fe^2^^+^, Cu^+^) under the formation of highly reactive hydroxyl radicals and/or perferryl species [102,103]. The probability for their formation rises at an enhanced H_2_O_2_ level and in the presence of free metal ions. In order to avoid dangerous reactions of metal ions, they are tightly controlled by several mechanisms in cells and tissues [104,105]. In airway lining fluids, transferrin is the major component for binding iron ions [29]. In secretions, lactoferrin is also able to scavenge and inactivate transition free iron ions [28].

## 5. Heme Peroxidases at Inflamed Epithelia

### 5.1. Major Functions of MPO and EPO at Inflamed Loci

With invading neutrophils, and to a minor degree with monocytes, MPO is attracted to inflamed loci. Several major functions of MPO are discussed [10]. This enzyme plays an active role during the phagocytosis of pathogens by neutrophils. It is apparently involved in the rapid pH increase in newly formed phagosomes and thus, provides optimal conditions for the action of serine proteases and microbicidal proteins.

In undergoing neutrophils, MPO is essential for the formation of neutrophil extracellular traps [106], where MPO and other proteins from neutrophils are tightly associated with DNA [107,108]. Traps are important for the defense against hyphenated fungi and microbes, independent of phagocytosis [109,110].

In addition, MPO released from neutrophils attaches to negatively charged surface areas, and forms complexes with several acidic proteins and polymers [44,111,112,113,114,115,116,117]. It is able to penetrate into endothelial cells. Moreover, after residing at the basolateral side, it affects the bioavailability of nitrogen monoxide in blood vessels [118,119]. On the basis of these findings, an involvement of MPO is discussed in the pathogenesis of numerous disease scenarios including atherosclerosis, vasculitis, rheumatoid arthritis, neurodegenerative diseases, etc. [120,121,122].

Attachment of MPO to cell surfaces at inflamed loci can induce the formation of antibodies against this protein [123]. The so-called myeloperoxidase-antineutrophil cytoplasmic antibodies play a role in the pathogenesis of different forms of vasculitis such as glomerulonephritis [124] and vasculitis of the upper and lower respiratory tract [125].

In reactions of type 2 immune response, eosinophils are recruited and activated together with mast cells and basophils [33]. Eosinophils are involved in the inactivation and killing of larger pathogens such as helminths and other parasites [126,127]. Moreover, they exhibit pronounced antimicrobial, antiviral, and antifungal activities [128]. After contact with pathogens, they release highly cationic proteins from their granules including EPO. In targeted cells, granule proteins create toxic pores, exhibit antiviral activities, and promote oxidative stress [9]. EPO contributes to the damage of the reactions via formation of HOBr [129]. Furthermore, these cells are important mediators of allergic diseases [130,131].

Similar to neutrophils, eosinophils are also known to release DNA-containing extracellular traps, the so-called eosinophil extracellular traps [132,133]. In contrast to neutrophils, the underlying process of trap formation in eosinophils is accompanied by the release of free extracellular granules that can target conidia from *Aspergillus fumigatus*, a fungus that is very common in allergic bronchopulmonary mycoses [134,135].

### 5.2. Disturbed Ion Transport in Cystic Fibrosis

In cystic fibrosis (CF), a multiorgan disease, secretion of anions from epithelial cells is disturbed as a result of genetic defects in CFTR. Mutations of the CFTR gene are classified into six main categories according to their impact on the CFTR protein. Severe CF phenotype is developed in people who are homozygous for class I, II, and III mutations [136]. In class I mutation, translation of the CFTR protein is prematurely terminated. Protein misfolding and enhanced proteasome degradation are key characteristics for class II mutations. The CFTR protein bearing class III mutation is incorporated into the plasma membrane, but is defective and exists most likely in a closed conformation. In the less severe phenotypes of classes IV to VI, functions of the anion channel are only partly disturbed. In many CF reviews, the type of CFTR mutations is not specified or CF is related to the most common ΔF508 mutation [137] belonging to the class II mutation.

The apical ion channel CFTR transports SCN^−^, chloride, bicarbonate, glutathione, and other anions into the mucous lining [138,139,140]. Together with the epithelial sodium channel and other proteins, the CFTR channel contributes to the regulation of volume and composition of extraepithelial fluid. In airway CF epithelia, reduced Cl^−^ and bicarbonate secretion and increased compensatory absorption of Na^+^ are responsible for water loss in the periciliary layer, enhanced adherence of mucins to epithelial cells, altered mucin properties, and decreased ciliary activity [141].

Moreover, it has been assumed that CF is associated with a lower SCN^−^ level in mucous linings and thus, with a decreased ability to generate the microbicidal ^−^OSCN/HOSCN [142]. Indeed, in cell culture experiments with airway epithelial cells that are defective in the CFTR channel, the ability to kill pathogens by ^−^OSCN/HOSCN was impaired [139,142]. However, the analysis of nasal airway surface liquids of persons with or without CF revealed no differences in SCN^−^ concentrations between both groups [143]. Evidently, the anion transporter pendrin can compensate deficient SCN^−^ transport by CFTR. Furthermore, pendrin is upregulated in airway epithelial cells by pro-inflammatory cytokines such as interleukin-4 [83,144,145].

On the contrary, the SCN^−^ concentration in nasal airway surface liquids was about 30 times higher than the SCN^−^ level in the serum of both the CF patients and healthy individuals [143]. In airway surface liquids, a similar ratio of 30 was determined for the SCN^−^ serum values of both the CF and healthy individuals [146,147].

### 5.3. Alterations in Mucous Properties in Cystic Fibrosis

In inflamed airway mucous fluids of CF patients, a lower pH of 6.8 was found as opposed to the unperturbed linings with a pH of 7.1 [148]. Moreover, protons were enriched in the nasal airway fluid of CF subjects with a pH of 6.57 in contrast to normal individuals with a pH of 7.18 [149]. In another study, equilibrium pH values of freshly excised sinonasal epithelia of CF patients were 7.08 as opposed to healthy subjects with a pH of 7.34 [150].

In the affected CF airways, a thicker and more viscous mucous was present that can result in mucous stasis, dilation of gland ducts, and reduced mucous clearance [151,152]. For the observed diminished mucociliary transport in CF, the effect of mucous concentration is more important than airway acidification [152]. Several factors are associated with the formation of a more compact and viscous mucous in CF patients. The diminished bicarbonate transport favors a more acidic pH [150,153,154] and higher Ca^2^^+^ concentration in the airway surface liquid [155]. A decrease in the pH enhances electrostatic interactions between mucins and thus, affects mucous viscosity [156]. The elevated Ca^2^^+^ mediates a tighter cross-linking of mucin strands most likely via binding to specific domains [155,157].

In CF patients, airway surface fluids are more susceptible due to their increased viscosity to different opportunistic bacteria and fungi. Lung infections by *Pseudomonas aeruginosa*, *Burkholderia cepacia*, and *Aspergillus fumigatus* are very common in CF [136,158]. These pathogens affect predominantly immunocompromised persons. In CF, a decrease in the pH of airway fluids contributes also to impaired bacterial killing [159].

Since early childhood, CF patients suffer from recurrent infections in lungs and many other organs. Deficiency in CFTR favors pro-inflammatory conditions including injury of epithelial cells. This damage can be further promoted by recruited immune cells, most of all by agents released from invading neutrophils as evidenced by the presence of elastase [160,161], MPO, and MPO products such as methionine sulfoxide in airway linings [162]. In severe cases, a progressive bronchiectasis can cause death by respiratory failure.

Moreover, the presence of oxidants contributes to more compact mucin structures by inducing additional disulfide cross-links [163]. Oxidants generated by MPO from invading neutrophils are known to contribute to mucous alterations. In addition, in the sputum of CF patients, a higher yield of 3-chlorotyrosine and other tyrosine oxidation products was detected [164].

### 5.4. Formation of HOCl by MPO in Inflamed Mucous Layers

Similar to LPO, MPO is also able to oxidize SCN^−^. Considering (pseudo)halide concentrations in blood, it has been found that MPO oxidizes at 0.1 M Cl^−^ and 0.1 mM SCN^−^ at a pH of 7, which is nearly the same amount as these anions [165]. In mucous linings, oxidation of SCN^−^ by MPO dominates due to the markedly higher SCN^−^ concentrations (see Section 4.4). Moreover, SCN^−^ is rapidly oxidized by HOCl and competes efficiently with other targets for HOCl [166]. Therefore, at a first glance, the MPO-mediated formation of HOCl seems to be unlikely in mucous linings. The preference of SCN^−^ over Cl^−^ in reactions with MPO Complex I at pH 7 is attenuated with the decreasing pH. At pH 5, the second order rate constant for the reactions of MPO Complex I with SCN^−^ in comparison to Cl^−^ is only 20 times higher in contrast to a ratio of nearly 400 at pH 7 [47].

In mucous linings and epithelia, pH data were usually evaluated by pH microelectrodes or pH-sensitive fluorophors in droplets taken from biopsies. In particular, in the case of inflamed materials, these kinds of measurements do not consider any local pH deviations that might result from the formation of microcompartments within the more compact mucous layer. Moreover, in the inflamed mucous of CF patients, DNA is known to be complexed with mucins [163]. Similar to negatively charged polyelectrolyte films [167], the more dense structure of mucous polymers and DNA can locally enhance the negative charges of these polymers and thus, enrich protons and other cations at local areas within the inflamed mucous layer. Interestingly, horseradish peroxidase and glucose oxidase complexed with DNA exhibit a higher activity resulting from a significant decrease of pH near the DNA surface [168].

The presence of MPO and MPO products was reported by several authors in the inflamed mucous of CF patients [162,164,169,170]. The cationic MPO resides predominantly at acidic loci [44,111,112,113,114,115,116,117]. Myeloperoxidase-DNA complexes are well known from neutrophil extracellular traps [106,107,108]. It is likely that undergoing neutrophils release traps at inflamed mucous surfaces.

The attraction of MPO to mucous polymers and the possibility of more acidic pH values at local areas would favor the formation of HOCl by the activated MPO. The ability of MPO to generate free HOCl rises considerably below pH 6 [171,172].

### 5.5. Allergic Inflammations

Different allergic diseases are closely associated with the unbalanced type 2 immune response, namely allergic asthma, atopic dermatitis, food allergies, hay fever, etc. Although the pathogenesis of asthma and related allergic diseases is very complex and far from thorough understanding, enhanced oxidation of SCN^−^ by heme peroxidases and, in particular, activation of EPO are under discussion to contribute to disease development.

An overproduction of ^−^OSCN/HOSCN has been assumed to play a role in the development of allergic inflammation in the lung, as shown in asthma model mice and asthma patients [173,174]. Upregulation of pendrin by IL-13 [145] and enhanced activity of heme peroxidases are responsible for this overproduction [173]. In airway epithelial cells, enhanced HOSCN activates via protein kinase A NFκB. At higher doses, HOSCN induces epithelial cell necrosis [175]. The release of IL-33 and other mediators from necrotic epithelial cells triggers an inflammatory response in airways [176,177]. Importantly, IL33 induces eosinophilia, and promotes several functions of eosinophils [178]. A vicious circle under participation of IL-13, IL-33, and ^−^OSCN may exaggerate and prolong the type 2 immune response in allergic diseases [175].

In SCN^−^ oxidation, both MPO and more efficiently EPO, produce in addition to the major product OSCN^−^, cyanate (^−^OCN) as a minor product [179]. The latter agent, which is in equilibrium with urea, promotes carbamylation of proteins, a condition that markedly affects the function of proteins and favors endothelial dysfunction and pro-inflammatory processes [180,181,182]. Although different amino acid residues can be modified in this way, lysine residues are a preferred target for carbamylation with the formation of homocitrulline moieties. At sites of eosinophilic inflammations, an increased number of carbamylated proteins was detected [183].

In addition, activation of EPO favors bromination and nitration of target molecules. The 3-bromotyrosine, 3,5-dibromotyrosine, as well as 3-nitrotyrosine residues were detected in proteins of the airway epithelium of patients with asthma [184,185,186].

### 5.6. Tissue Damage by Heme Peroxidases and Their Products

During an inflammatory response, additional tissue damage can occur by agents released from the activated immune cells and undergoing tissue cells. Under chronic inflammatory conditions, the repeated release of DAMPs from necrotic tissue cells can frequently foment the inflammatory process [30]. In other words, inflammation is not terminated and tissue homeostasis is not restored adequately. Immunocompromised individuals are most of all affected by recurrent inflammations and opportunistic infections [3,187,188]. In the host’s tissues, there is a tight balance between the damage by agents from immune and necrotic cells and the already existing mechanisms to resist and inactivate these destructive agents [1]. A shift in this balance towards damaging processes due to insufficiency or exhaustion of the host’s defense mechanisms favors chronic conditions and disease processes with long-lasting inflammations. Examples for the interplay between potentially damaging agents and antagonizing principles are given in reference [10].

The major focus will be directed here on the potential role of MPO and EPO and their products in the damage of epithelial cells of mucous surfaces under chronic inflammatory conditions. Activated neutrophils and eosinophils release potential cytotoxic agents at inflammatory loci. In addition to MPO and MPO-derived oxidants, neutrophils participate in destructive reactions with numerous proteolytic enzymes such as elastase, cathepsin G, proteinase 3, lysozyme, collagenase, and gelatinase and the formation of superoxide anion radicals and H_2_O_2_ [10]. Cytotoxic eosinophil agents include EPO and EPO products, major basic proteins, eosinophil-derived neurotoxins, eosinophil cationic proteins, superoxide anion radicals, and H_2_O_2_ [9]. Therefore, heme peroxidases do not act alone in tissue damage, but in concert with other destructive components. It is very challenging to predict which agents predominate preferentially in damaging reactions. This highly depends on the individual status of antagonizing components in the host’s cells and tissues.

In mucous linings, another set of antagonizing agents is present that can deactivate toxic components in comparison with other body fluids. In blood, MPO released from neutrophils is antagonized by the plasma protein ceruloplasmin, forming a tight inhibitory complex with MPO [189,190,191,192,193]. Moreover, ceruloplasmin forms an inhibitory complex with EPO [193]. Only minor amounts of ceruloplasmin were detected in the airway lining fluid [194].

Heme peroxidases can contribute to the formation of hypohalous acids and hypothiocyanite in order to damage the reactions. In mucous lining fluids, glutathione deactivates these halogenated species and functions as a cofactor for extracellular glutathione peroxidase to reduce H_2_O_2_ [26,101]. In healthy individuals, glutathione levels of the airway lining fluid are more than 100-fold higher than in the plasma [26]. In addition to antioxidative activities, this extracellular glutathione pool is a reservoir of cysteine for the synthesis processes in epithelial cells [195]. Several pathological conditions are known with a reduced level of glutathione in airway mucous linings, namely CF and idiopathic pulmonary fibrosis [196,197]. Therefore, a reduced level of extracellular glutathione attenuates protection against HOCl and HOBr, and worsens the control over ^−^OSCN/HOSCN and H_2_O_2_.

## 6. Conclusions

As important components of immune reactions, the heme peroxidases LPO, MPO, and EPO and their products contribute to the protection against pathogens. On the other hand, they can be involved in different pathologies concerning secretory mucous epithelia. These opposite activities of heme peroxidases reflect the general situation of immune defense. Aggressive metabolites have to be used to control and combat against pathogens. Under certain conditions, these metabolites can be directed against the host’s own cells and tissues.

Mucous epithelial cells secrete a variety of immune-protective agents, namely major agents LPO, H_2_O_2_, and SCN^−^, which produce bactericidal ^−^OSCN/HOSCN, as well as mucins, glutathione, and transferrin. In unperturbed mucous linings, secreted components maintain microorganisms under control and scavenge potential toxic agents. Alterations in the composition of mucous linings, as well as damage of epithelial cells promote the development of inflammatory processes. In CF, the secretion of anions such as chloride, glutathione, bicarbonate, and SCN^−^ is disturbed due to the defective CFTR channels. Therefore, serious alterations in mucous properties result, which can favor colonization of microbes on mucous surfaces. On the other hand, damage of epithelial cells of mucous surfaces is associated with the induction of type 2 immune response, including the recruitment and activation of eosinophils. Moreover, overproduction of ^−^OSCN/HOSCN, carbamylation of proteins, and appearance of brominated and nitrated amino acid residues can contribute to epithelial cell necrosis. The latter mechanisms are discussed in the pathogenesis of allergies such as asthma.

At inflammatory sites, invading leukocytes release MPO and EPO. Unlike LPO, these two heme peroxidases are able to produce the powerful oxidants HOCl and HOBr. In inflamed mucous linings, the chlorinating activity of MPO is apparently favored by complexes of MPO with DNA, the resulting decrease of local pH, by glutathione deficiency, and when competing effects by SCN^−^ are limited. Further research is highly necessary to verify these preliminary conclusions. Similar corollaries are valid for the participation of EPO and EPO products in disease progression at mucous surfaces.

## Figures and Tables

**Figure 1 antioxidants-10-01805-f001:**
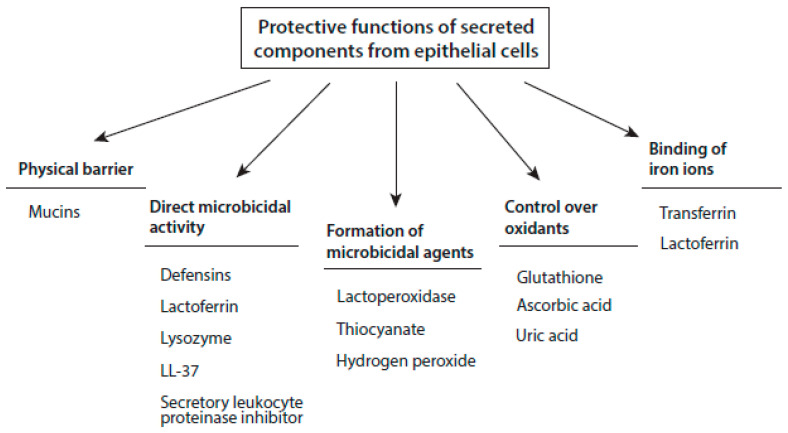
Major immune-protective functions of agents secreted from epithelial cells of mucous surfaces.

**Figure 2 antioxidants-10-01805-f002:**
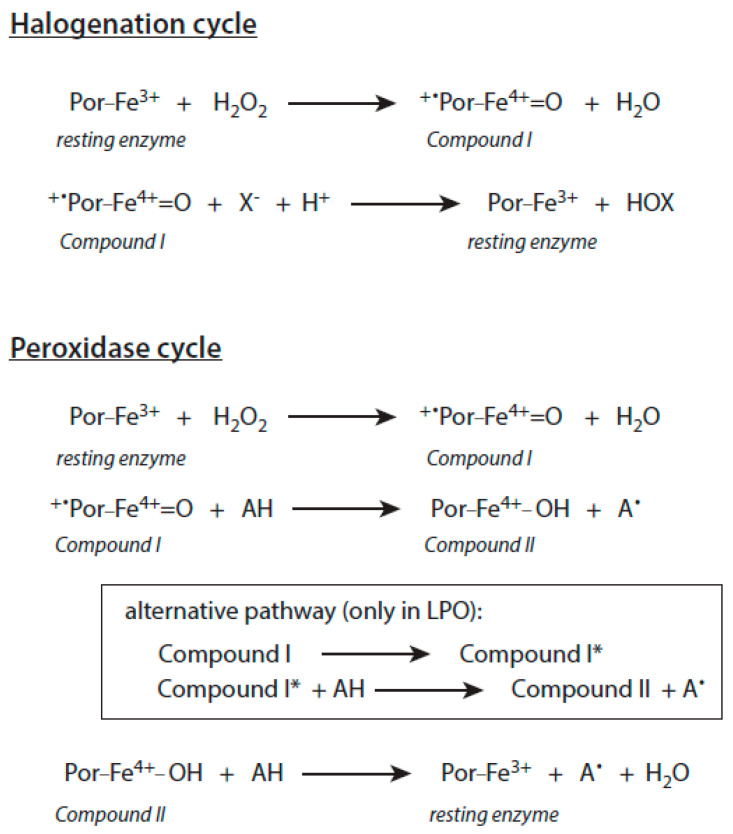
Halogenation and peroxidase cycles of heme peroxidases. Further explanations are given in the text. Por-Fe denotes the porphyrin-iron complex. X^−^ stands for Cl^−^, Br^−^, I^−^, and SCN^−^. HOX is the corresponding (pseudo)hypohalous acid. AH is an oxidizable substrate, and A^∙^ the resulting substrate radical.

**Table 1 antioxidants-10-01805-t001:** Selected structural properties of heme peroxidases.

Property	MPO	EPO	LPO
Overall structure	dimeric [40]	monomeric [38]	monomeric [39]
Molecular weight	144 kDa [40]	71 kDa [38]	77.5 kDa [11], Faraji 2017
*p*-value	>10 [42]	>11 [43]	9.5 [41]
Number of linkages between heme and apoprotein	3 [46]	2 [45]	2 [39]
Heme bending	stronger as in EPO and LPO	slight	slight
Displacement of central heme iron to the proximal side	0.2 Å [46]	unknown	0.1 Å [39]
Soret band location	430 nm [47]	413 nm [48]	412 nm [49]

## Data Availability

Data is contained within the article.

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
