# Peer review of "Heme Peroxidases at Unperturbed and Inflamed Mucous Surfaces"

_antioxidants, 2021, doi:10.3390/antiox10111805_

Round 1

Reviewer 1 Report

This is a well-written review on the role of LPO, MPO, and EPO in especially mucosal host defense and inflammation by an author with considerable experience in the chemistry and biology of these heme peroxidases. This is well balanced in regards to the beneficial and harmful functions of these enzymes.

I have only minor remarks.

When abbreviations are first used in the text, please give the full name first.

  1. Line 169: ‘hydrogen peroxide’ should be abbreviated to ‘hydrogen peroxide (H2O2)’, and ‘hydrogen peroxide’ shown in lines 201, 238, 243, 278, 284,310, 494, and 496 should be replaced to ‘H2O2.’
  2. Lines 179 and 189: ‘hypothiocyanite’ and ‘thiocyanate’ should be abbreviated to ‘hypothiocyanite (OSCN-)’ and ‘thiocyanate (SCN-)’ respectively, and those full names shown in lines 239, 242, 244, 251, 260, 263, 464, and 507 should be replaced to their abbreviated forms. ‘Cyanate’ in line 464 should be ‘-OCN.’
  3. Lines 272 and 380: ‘interleukin-4’ should be IL-4.
  4. Lines 217, 224, 230, 318, 345, 416, 417, and 440: ‘myeloperoxidase’, ‘eosinophil peroxidase’, and ‘lactoperoxidase’ should be ‘MPO’, ‘EPO’, and ‘LPO’, respectively.
  5. Lines 346, 365, and 529: ‘hypobromous acid’ and ‘chloride’ could be ‘HOBr’ and ‘Cl-’, respectively.

Other minor remarks.

  1. Lines 283 and 781: ‘xanthin’ could be ‘xanthine’
  2. Line 459: ‘IL33’ could be ‘IL-33’
  3. Line 519: ‘one’ could be ‘on’

Author Response

Response to reviewer 1

 Thank you very much for your review. The following changes have been performed according to your suggestion.

Full names and abbreviations are used in the revised version as you proposed.

The three other minor remarks were corrected accordingly.

Reviewer 2 Report

The introduction describes the multiple defensive mechanisms that the body provides including the roles of many intriguing molecules such as cryptdins. It would be fascinating to know more about the structure and function of these numerous compounds and the review goes a long way to filling this knowledge gap. Perhaps the author could outline some more of this information briefly. It was very rewarding to find that structural information is presented on the heme peroxidase family of defense proteins. 

The heme bending which is described covers a fascinating effect. It would be very interesting to know more about the structural techniques used to establish this phenomenon. 

The generation of hypohalous compounds is indeed a fascinating effect. This section mentions the role of nitrite in redox and defensive pathways and it would be fascinating to learn more of the roles of this molecule. 

The review rightly emphasises how the roles of enzymes in defense processes is corroborated by knock-down studies. I am intrigued that pseudo-halides have a key biological role and this is something that many will not have heard of before and is something which the review makes many steps to remedy. 

The review moves on to cover cystic fibrosis - a severe illness which presents many challenges to the study of inflamatory disease. The exaggerated imune response in allergy is succinctly described. 

Above all this review is very interesting resumé of current knowledge and understanding of inflammatory process at mucosal surfaces and I have no hesitation in its recommending publication. 

I am very impressed that the author has made so much effort to provide a comprehensive and authoritative review of this fascinating and highly topical subject. 

Author Response

Response to reviewer 2

 Thank you very much for your review. The following changes have been performed according to your suggestion.

Concerning the heme bending, I added a short remark (lines 157 and 158) that these data were obtained from X-ray data in case of MPO and LPO, and from comparative biochemical analysis for EPO.

Yes, I agree with you that some more information about organization of immune defense at mucous surfaces would be useful. My intention was, to show that several routes for defense reactions exists in mucous lining fluids and secretions including different microbicidal agents and of course the lactoperoxidase system. As the main topic of this review is the role of heme peroxidases at unperturbed and inflamed mucous surfaces, I did not further extend this section.

Biochemical reactions of heme peroxidases are very complicated. Besides reactions of the halogenation cycles, many substrates were oxidizes in the peroxidase cycle including nitrite. Oxidation of nitrite by MPO is described in detail in reference 55.
